# Prevalence, Trends, and Drivers of the Utilization of Unskilled Birth Attendants during Democratic Governance in Nigeria from 1999 to 2018

**DOI:** 10.3390/ijerph17010372

**Published:** 2020-01-06

**Authors:** Felix Akpojene Ogbo, Felicity F. Trinh, Kedir Y. Ahmed, Praween Senanayake, Abdon G. Rwabilimbo, Noel E. Uwaibi, Kingsley E. Agho

**Affiliations:** 1Translational Health Research Institute, School of Medicine, Western Sydney University, Campbelltown Campus, Locked Bag 1797, Penrith, NSW 2571, Australia; K.Ahmed@westernsydney.edu.au (K.Y.A.); praween.senanayake@gmail.com (P.S.); rwabi1977@gmail.com (A.G.R.); k.agho@westernsydney.edu.au (K.E.A.); 2General Practice Unit, Prescot Specialist Medical Centre, Welfare Quarters, Makurdi 972261, Benue State, Nigeria; 3Neeta City Medical Dental and Specialists Centre, Suite 1, Level 2/54 Smart St, Fairfield, NSW 2165, Australia; ftri0660@uni.sydney.edu.au; 4College of Medicine and Health Sciences, Samara University, P.O. Box 132 Samara, Ethiopia; 5Chato District Council, Chato, Geita Region, Northwestern, Tanzania; 6Edo University Iyamho, Kilometer 7 Auchi–Abuja Expressway, Auchi, Edo State, Nigeria; noel.uwaibi@gmail.com

**Keywords:** traditional birth attendants, other unskilled birth attendants, home birthing, maternal health, Nigeria

## Abstract

Comprehensive epidemiological data on prevalence, trends, and determinants of the use of unskilled birth attendants (traditional birth attendants (TBAs) and other unskilled birth attendants) are essential to policy decision-makers and health practitioners, to guide efforts and resource allocation. This study investigated the prevalence, trends, and drivers of the utilization of unskilled birth attendants during democratic governance in Nigeria from 1999 to 2018. The study used the Nigeria Demographic and Health Surveys data for the years 1999 (n = 3552), 2003 (n = 6029), 2008 (n = 28,647), 2013 (n = 31,482), and 2018 (34,193). Multivariate multinomial logistic regression was used to investigate the association between socioeconomic, demographic, health-service, and community-level factors with the utilization of TBAs and other unskilled birth attendants in Nigeria. Between 1999 and 2018, the study showed that the prevalence of TBA-assisted delivery remained unchanged (20.7%; 95% CI: 18.0–23.7% in 1999 and 20.5%; 95% CI: 18.9–22.1% in 2018). The prevalence of other-unskilled-birth-attendant use declined significantly from 45.5% (95% CI: 41.1–49.7%) in 2003 to 36.2% (95% CI: 34.5–38.0%) in 2018. Higher parental education, maternal employment, belonging to rich households, higher maternal age (35–49 years), frequent antenatal care (ANC) (≥4) visits, the proximity of health facilities, and female autonomy in households were associated with lower odds of unskilled birth attendants’ utilization. Rural residence, geopolitical region, lower maternal age (15–24 years), and higher birth interval (≥2 years) were associated with higher odds of unskilled-birth-attendant-assisted deliveries. Reducing births assisted by unskilled birth attendants in Nigeria would require prioritized and scaled-up maternal health efforts that target all women, especially those from low socioeconomic backgrounds, those who do not attend antenatal care, and/or those who reside in rural areas.

## 1. Introduction

In the last four decades, reducing maternal mortality has been consistently highlighted as an area for greater attention by global health organizations. This has been done through various efforts, including the Safe Motherhood Initiative [1,2] and the Millennium Development Goals [3], and more recently, the Sustainable Development Goals (SDGs) agenda [4]. SDG–3.1 aims to reduce global maternal mortality ratio (MMR) to less than 70 per 100,000 live births by 2030 [4]. Increasing women’s access to comprehensive antenatal care (ANC) and health facility birthing with skilled birth assistance are effective strategies to achieve SDG–3.1 [5,6,7]. Also, appropriate utilization of trained traditional birth attendants (TBAs), where required, can improve health outcomes for both women and babies [8,9]. A TBA “is a person, usually a woman, who assists the mother during childbirth and who initially acquired her skills by delivering babies herself or by working with other TBAs”. In contrast, a trained TBA “is a person who has received a short course training to improve her skills and knowledge through the modern health care sector” [8]. However, it may be challenging for community members to differentiate trained TBAs from untrained TBAs.

Globally, the MMR has declined by 38% from 342 deaths in 2000 to 211 deaths per 100,000 live births in 2017 [10,11]. However, the burden of MMR remains high, as about 830 maternal deaths occur daily worldwide, and over 99% of these deaths occur in low- and middle-income countries (LMICs) [5,11]. In Sub-Saharan African countries, the MMR has declined by 39% from 870 deaths in 2000 to 533 deaths per 100,000 live births in 2017 [10,11]. In these emerging nations, Nigeria ranked fourth among the top 10 African countries with the highest MMR, with an estimated 917 deaths per 100,000 births in 2017 [11].

In rural and remote areas of Nigeria, reducing the burden of maternal morbidity and mortality and improving birthing experiences of women would require increased access to skilled birth attendants [6]. This is important, as there is often a lack of skilled health practitioners in those settings [9], which may make women seek alternative health care, including assistance from unskilled birth attendants. However, the use of unskilled birth attendants (TBAs, relatives, or friends) can lead to considerable morbidity and disability, and even death of both the mother and baby. These adverse outcomes can occur because unskilled birth attendants usually lack the required knowledge and skills to risk-stratify or manage common pregnancy or childbirth complications, such as hemorrhage, eclampsia, and obstructed labor [8]. In Nigeria, births assisted by TBAs and other untrained personnel are common; however, epidemiological data on their prevalence, trends, and determinants are limited. Some studies suggest that the prevalence varied by regions, 50–57% in some parts of Southern Nigeria [12,13]. One of the studies indicated that culture, compassionate attitude of TBAs, non-availability of health facility, and distance from the health facility were associated with the use of unskilled birth attendants [12].

A recently published study in *The Lancet* has indicated that countries with democratic governance are more likely to have health gains from noncommunicable diseases (e.g., cardiovascular disease) compared to countries with autocracies [14]. Similar findings have been reported for infant and child mortality and life expectancy at birth globally [15,16,17,18]. Nigeria gained its independence from the United Kingdom in 1960. Until 1999, Nigeria was largely governed by military regimes [19], with limited government health spending per capita and developmental assistance for health [20,21]. Since the end of military rule in 1999, Nigeria has been conducting scheduled elections to select both national and subnational leaders. While there are still many areas for improvement in Nigeria’s democratic process [22], for the first time, in 2015, the incumbent presidential candidate conceded election defeat to the opposition candidate [23]. This action was significant in Nigeria’s history, as it provided a precedent for future elections and potentially improved political competition and reduced prospects for autocracies in the country. Additionally, the Nigeria government’s health spending has gradually increased since 1999 [21]. It is, however, uncertainty whether democratization and subsequent increase in health-care spending have translated into improvements in maternal health outcomes, including a decline in the use of unskilled birth attendants in Nigeria between 1999 and 2018.

To the authors’ knowledge, no nationally representative studies have investigated the prevalence, trends, and drivers of the utilization of unskilled birth attendants during the democratic rule in Nigeria from 1999 to 2018, nor has there been an analysis of the most recent national household data, the Nigeria Demographic and Health Survey Data (NDHS), on the use of unskilled birth attendants. These recent data potentially reflect the current socioeconomic, demographic, health, and political context of Nigeria. Detailed information on the levels, trends, and determining factors of unskilled birth attendants’ use among Nigerian women is essential to maternal health advocates, health practitioners, and policymakers. This information will be helpful to stakeholders in national resource allocation and priority setting. This study aims to examine the prevalence, trends, and drivers of the utilization of unskilled birth attendants during the period of democratic governance in Nigeria from 1999 to 2018.

## 2. Materials and Methods

### 2.1. Data Sources

The study used the NDHS datasets for the period spanning 1999 to 2018. The data were collected by the National Population Commission (NPC), with technical assistance received from Inner City Fund (ICF) International, USA. The NDHS aims to provide relevant maternal- and child-health indicators (such as antenatal care, delivery care, postnatal care, and infant and young child feeding) to facilitate planning, implementation, monitoring, and evaluation. A two-stage stratified sampling technique was used to capture the statistical snapshot of the country. In stage one, enumeration areas (EAs) were selected proportional to the household size of administrative areas. In stage two, fixed number of households were selected from each EA to select the study participants (n = 3552 in 1999, n = 6029 in 2003, n = 28,647 in 2009, n = 31,482 in 2013, and 34,193 in 2018). The number of participants per survey increased over the 19 years, due to population growth and improved data-collection methods [24]. Using standardized questionnaires, maternal health information (including birth-assistance data), as well as sociodemographic and economic characteristics, was collected from eligible women aged 15–49 years. Women who were residents or visitors in the households 24 h before the survey were selected and eligible to participate, with greater than 95% response rates in the surveys. Detailed information on the survey methodology is provided in the final NDHS reports [24,25,26,27,28]. In the present study, a total weighted sample of 103,903 most recent live-birth children under 5 years of age was used to minimize the potential effect of recall bias, consistent with Measure DHS approach [24] and used in previously published studies [29,30,31,32].

### 2.2. Outcome Variables

TBAs and other-unskilled birth-attendant-assisted deliveries were the study outcome variables. TBA-assisted birth was defined as women who had a TBA present during delivery. Delivery assistance received from other untrained attendants was measured as births assisted by community health extension workers, relatives, friends, or no help at all [24]. We also report on the prevalence of skilled birth assistance, measured as births that occurred with the assistance of doctors, nurses/midwives, and auxiliary nurses/midwives, to leverage the comprehensiveness of the data [24]. Because of a lack of relevant data, this study does not seek to determine a formal relationship between policy implementation during democratic governance and the use of unskilled birth attendants but to investigate trends in unskilled birth attendants use within the context of democratization in Nigeria.

### 2.3. Study Variables

The study variables were selected based on evidence from LMICs [12,33,34,35] and data availability, and were broadly categorized into four groups using an adopted Anderson behavioral model (Figure 1) [36]. These included community-level factors, predisposing factors (sociodemographic and media exposure), enabling factors (health service factors and women empowerment factors), and need factors.

The community-level factors included the place of residence (grouped as ‘urban’ and ‘rural’) and geopolitical region (‘North-Central’, ‘Northeast’, ‘Northwest’, ‘Southeast’, ‘Southwest’, and ‘South-South’). We considered North-Central Nigeria as the referenced category because it is the first region on the list of Nigeria’s geopolitical regions, as described in the NDHS reports. The sociodemographic factors included maternal age at delivery, maternal and paternal education, maternal employment, and household wealth status. Maternal age at delivery grouped as ‘15–24 years’, ‘25–34 years’, and ‘35–49 years’. Maternal and paternal education were grouped as ‘no schooling’, ‘primary education’, and ‘secondary or higher education’. Maternal employment was grouped as ‘no employment’, ‘formal employment’, and ‘informal employment’. Formal employment included women who were working in professional, technical, managerial, clerical, and services areas. Women who were working in agriculture and manual works were grouped under informal employment; and those who were not working were grouped under no employment [21]. The NDHS used principal components analysis to calculate the household wealth index based on a series of variables relating to ownership of household assets, such as cattle and bicycles; type of materials used for housing construction; and types of water source and sanitation facilities. The NDHS categorized household wealth index into five quintiles (poorest, poorer, poor, rich, or richest). In this study, the household wealth index was reclassified as ‘poor’, ‘middle’, or ‘rich’, to increase the sample within each category, consistent with past studies [31,37,38]. Media exposure included frequency of reading magazines or newspapers, frequency of listening to the radio, and frequency of watching TV, which were grouped as ‘Yes’ or ‘No’.

The enabling factors included health service factors and women empowerment factors. Health service factors included frequency of ANC visits (grouped as ‘none’, ‘1–3 visits’, and ‘4 or more visits’) and distance from the health facility (grouped as ‘a big problem’ and ‘not a big problem’). Women empowerment factors included the decision to seek medical help, the power to make household purchases, and requiring someone to accompany the woman during medical visits, which were grouped as ‘a big problem’ and ‘not a big problem’. The need factor included the desire for pregnancy, which was grouped as ‘desire for pregnancy’ or ‘no desire for pregnancy’. In the present study, we selected the reference categories for each study variable, based on past studies and/or specific research question relating to each variable [38,39,40,41].

### 2.4. Statistical Analysis

Preliminary analyses were conducted to describe the characteristics of the study participants by calculating the frequencies and percentages of the study factors. This was followed by calculating the prevalence of skilled birth attendants, TBAs, and other unskilled birth attendants’ delivery and by the study factors (community-level, predisposing, enabling, and need factors). Point percentage change with 95% confidence intervals (CIs) in the prevalence of TBAs and other unskilled birth attendants’ delivery was calculated to show the difference in the prevalence of TBAs and other-unskilled-birth-attendant deliveries within each study variable and over the study period from 1999 to 2018.

A multivariate multinomial logistic regression model was used to examine the association between the study factors and the outcome variables by considering the skilled assisted delivery as the reference category. A four-staged model was employed in the multivariate analyses, similar to the adopted conceptual model described by Andersen [23] and used in past studies [19,24]. In stage one, community-level factors were entered into the model, after controlling potential confounders (predisposing, enabling, and need factors). In stage two, predisposing factors were entered to the model, after adjusting for community-level, enabling, and need factors. In stage three and four, the same analytical strategy was used for the enabling and need factors in the third and fourth stages, respectively. Any collinearity between study factors was also investigated, where applicable, but none was evident in the analyses.

In this study, we used the combined dataset to increase the statistical power to detect any association between the study factors and the outcomes, as well as to examine any driver of change in TBA and other unskilled birth attendants’ deliveries over the study period. In models of the combined data, adjustment for year of the survey and cluster was done to account for the increased sample size and cluster in the dataset. We also estimated P for trend in models of the combined data to determine changes within each study factors over time. Odds ratios (ORs) with 95% CIs were calculated as the measure of association between the study factors and outcome variables. All analyses were conducted in Stata version 14.0 (Stata Corp, College Station, TX, USA), with ‘svy’ command used to adjust for sampling weights, clustering, and stratification effects, with ‘lincom’ command for estimating percentage points changes, and ‘mlogit’ function for multinomial models [25].

### 2.5. Ethics

The survey protocols were reviewed and approved by the National Health Research Ethics Committee of Nigeria (NHREC) and the ICF Institutional Review Board before the surveys were conducted, with informed consent also obtained from respondents before the data were collected. The lead author sought approval from Measure DHS to use the data, and permission was granted.

## 3. Results

### 3.1. Prevalence of Delivery Assisted by TBAs and Other Unskilled Birth Attendants by Study Factors

Between 1999 and 2018, the highest prevalence of TBA-assisted delivery was observed among women who resided in Northwest Nigeria (32.0%); followed by those resided in the Southwestern region (29.6%), while the lowest prevalence was in women who resided in the Southeastern region (6.4%) (Table 1). In the same period, the highest percentage point increase in the prevalence of TBA-assisted delivery was observed among women who resided in Southwest Nigeria (percentage change = 20.6; 95% CI: 15.2–25.9), followed by those who had attained primary education (Percentage change = 7.6; 95% CI: 3.3–11.8) and those whose partners had attained primary education (Percentage change = 7.6; 95% CI: 3.4–11.8) (Table 1).

Women who did not attend schooling had the highest prevalence of delivery assisted by other untrained individuals (58.0%), followed by those who resided in poor households (56.5%), while the lowest prevalence was observed among women who read magazines or newspapers (10.6%) (Table 2). Women who did not have an ANC visit had the highest decline in the prevalence of other-unskilled-birth-attendant-assisted deliveries (percentage change = −18.4 95% CI: −23.8, −12.9) (Table 2).

### 3.2. Trends in Delivery Assistance in Nigeria, 1999–2018

Between 1999 and 2018, the study showed that the prevalence of TBA-assisted delivery remained relatively unchanged (20.7%; 95% CI: 18.0–23.7% in 1999 and 20.5% 95% CI: 18.9–22.1% in 2018; *p*-trend = 0.912). The prevalence of other unskilled birth attendants’ utilization significantly declined from 45.5% (95% CI: 41.1–49.7%) in 2003 to 36.2% (95% CI: 34.5–38.0%) in 2018; *p*-trend < 0.001. This study also showed some improvements in the use of skilled birth attendants among Nigerian women from 38.1% (95% CI: 18.9–22.1%) in 2013 to 43.3% (95% CI: 18.9–22.1%) in 2018; *p*-trend = 0.492 (Figure 2).

### 3.3. Determinants of TBA-Assisted Delivery

Between 1999 to 2018, women who lived in rural households had higher odds of TBA-assisted deliveries compared to those who were in urban households. In all geopolitical regions of Nigeria, women had higher odds of being assisted by TBAs compared to their counterparts in the reference group (North-Central) (Table 3). The likelihood of using TBAs was lower among women who had attained secondary or higher education compared to those who had no schooling. Women whose husbands attained secondary or higher education were less likely to utilize TBAs during delivery compared to those whose husbands had no schooling. The likelihood of using TBAs during childbirth was lower among women who were in formal employment compared to those who were not in employment (Table 3). Women who resided in middle and rich households had lower odds of TBA-assisted delivery compared to those who resided in poor households. Older women (35–49 years) were less likely to be assisted by TBAs, while younger women (15–24 years) were more likely to be assisted by TBAs compared to those within the age group of 25–34 years. Women who reported a birth interval of 24 months or more had higher odds of TBA-assisted delivery compared to those who had no previous birth (Table 3). Women who read magazine or newspaper and/or watched television had lower odds of TBA-assisted delivery compared to their counterparts. Women who had frequent ANC (≥4) visits were less likely to be assisted by TBAs compared to those who had no ANC visits. Women who reported distance from the health facility, those who had money to pay for health services, or who required someone to accompany them to a health facility were less likely to use TBAs during delivery compared to their counterparts (Table 3).

### 3.4. Determinants of Other-Unskilled-Birth-Attendant-Assisted Delivery

Over the study period, women who resided in rural households had higher odds of using other unskilled birth attendants compared to those who lived in urban households. In Northeast and Northwest Nigeria, women had higher odds of being assisted by other unskilled birth attendants compared to their counterparts in North-Central Nigeria (Table 4). Women who resided in Southeast and Southwest Nigeria were less likely to utilize other unskilled birth attendants compared to those who resided in North-Central Nigeria. The odds of using other unskilled birth attendants during delivery was lower among women who attended secondary or higher education compared to those who had no schooling (Table 4). Women whose husbands attained secondary or higher education were less likely to utilize other unskilled birth attendants during delivery compared to those whose husbands had no schooling. Women who resided in rich households were less likely to be assisted by other unskilled birth attendants compared to those who resided in poor households. Young women (15–24 years) had higher odds of using other unskilled birth attendants during delivery compared to those who were in younger age groups (25–34 years) (Table 4). Women who had frequent ANC (≥4) visits were less likely to be assisted by other unskilled birth attendants compared to those who had no ANC visits. Women who reported a birth interval of 24 months or more had higher odds of other-unskilled-birth-attendant-assisted delivery compared to those who had no previous birth. Women who reported distance from the health facility, those who had money to pay for health services, or required someone to accompany them to the health facility were less likely to use other unskilled birth attendants during delivery compared to their counterparts (Table 4).

## 4. Discussion

Our study showed that the prevalence of TBA-assisted delivery remained unchanged over the study period (approximately 21.0% in both 1999 and 2018). The proportion of Nigerian women who received assistance from other unskilled birth attendants decreased between 2003 and 2018 (45.5% vs. 36.2%, respectively). This study also showed some improvements in the use of skilled birth attendants in Nigeria from 38.1% in 2013 to 43.3% in 2018. Higher parental education, maternal employment, belonging to rich households, reading magazines or newspapers, watching television, higher maternal age (35–49 years), frequent ANC (≥4) visits, the proximity of health facilities, and having money to pay for health services were associated with lower odds of TBA- and other-unskilled-birth-attendant-assisted births. In contrast, northeast, northwest, and rural residence and higher birth interval (≥2 years) were associated with higher odds of TBAs and other-unskilled-birth attendant-assisted deliveries. Residence in all Nigeria’s geopolitical regions was associated with increased odds of TBA use, while residence in Southern Nigeria decreased the likelihood to use other unskilled birth attendants during delivery.

The increased use of skilled birth attendants between 2013 and 2018 and a reduction in other unskilled birth attendants’ use among Nigerian women over the same period are important contextual findings. Evidence shows that democratic governance is more likely to improve health outcomes compared to autocracy as a result of greater attention that is given to health promotion services, greater receptiveness to public feedback, improved government health spending, and protection of media freedom [42]. Within the context of democratic governance in Nigeria, the introduction of free maternal health policies in various Nigerian states may have resulted in the decline in the proportion of Nigerian women who received delivery assistance from other unskilled birth attendants [43,44,45]. Nevertheless, caution must be exercised in trying to explain the reason for why this decline has occurred, because there may be other unpublished interventions that were implemented at the national and subnational levels in Nigeria that may have contributed to the change in prevalence.

Our study showed that the use of TBAs during childbirth remained unchanged over the study period (1999–2018). The analyses also indicated that women in all geopolitical regions had higher odds of being assisted by TBAs, suggesting that the benefits of democratization may not have translated to limited use of TBAs. Nigeria is one of the most diverse countries in the world, with between 300 and 400 ethnic and sociocultural groups, depending on the classification, and over 400 languages [46,47,48]. While the use of TBAs has been associated with socioeconomic, demographic, and health-service factors in many Sub-Saharan African countries, cultural norms can also play an important role in the use of TBAs [8,33,49,50,51]. In these settings, residents perceived the role of both health-care professionals and TBAs as important in providing maternal-health-care services. Additionally, recent qualitative studies conducted in Nigeria [52] and Zambia [53] reported some factors that can also promote the use of TBAs over skilled birth attendants in maternal-health facilities. These factors included sociocultural norms, long waiting times in health facilities, substandard facilities, and the poor attitude of health workers and their inattention to women in labor. Some of these findings mirrored a previous study from Nigeria which reported that TBAs have a compassionate attitude toward women in labor [12].

In Nigeria, past studies have indicated that TBAs’ practices were unsafe and could lead to adverse maternal and child outcomes [54,55,56]. In supportive efforts to improve skilled birth attendants in Nigeria, Okonofua and Ogu [33] have argued that interventions that aim to reduce the use of unskilled birth attendants should focus on redirecting women to modern maternity-health facilities. However, in rural and remote communities of Nigeria, where the majority of women reside and there is often scarcity of skilled birth attendants associated with low cost of TBAs services and/or long distance to modern health facilities, the use of trained TBAs in those settings may be warranted in the short-term [8,9,57,58,59]. To increase the availability and accessibility of skilled birth attendants in Nigeria, health practitioners and policymakers must design context-specific maternal-health programs that also consider the use of trained TBAs to improve communication between TBAs and health facilities. Training for skilled birth attendants may also be considered in order to improve women birthing experiences.

We found that women who resided in the rural areas of Nigeria were more likely to utilize TBAs and other unskilled birth attendants during childbirth. This result potentially reflects the significant barriers (such as long-distance to health facilities, lack of transportation, and/or poorer socioeconomic status) that rural women face in accessing skilled birth attendants [12]. Jimoh et al. [60] found that, in the rural settings of Nigeria, traditional practices encouraged home delivery assisted by TBAs and also delayed the onset of breastfeeding. The author also suggested that residents in rural Nigeria adhere more to sociocultural norms compared to their urban counterparts, which may contribute to the use of unskilled birth attendants in those settings [60]. Meaningful paradigm shifts in maternal-health efforts need to occur in Nigeria’s rural communities, guided by culturally sensitive and targeted health education, galvanized with strong political will if the utilization of skilled birth attendants is to be increased. Further, incentivization of skilled birth attendants to work in rural and remote regions of Nigeria would also increase access for rural women to have health-professional birth attendants during delivery.

Empirical evidence has shown that empowering women with education improves their decision-making power, which can subsequently increase maternal-health-service utilization [61,62,63]. Our study found that women who attained secondary or higher education had lower odds of being assisted by TBAs or other unskilled birth attendants. This finding is consistent with previously published studies conducted in Ghana [64], Kenya [34], and Afghanistan [35], where higher maternal education was related to the use of skilled birth attendants. The inverse relationship between higher maternal education and unskilled-birth-attendant-assisted delivery could be that women with higher levels of education may be more responsive to health-promotion messages [65] and may have a greater insight into the increased safety measures associated with health facility birthing and use of skilled birth attendants [35]. It is also possible that educated women are less likely to be persuaded by cultural paradigms that may encourage the use of TBAs and/or other unskilled birth attendants. Furthermore, educated women are also likely to be employed, married to educated partners, and thus, have the resources to access health care facilities and skilled birth attendants during delivery. The analyses also showed that women whose husbands were educated and those who had autonomy over health-seeking were less likely to receive delivery assistance from unskilled birth attendants. Our findings suggest that the Nigeria government should strengthen education for girls, as articulated in the SDG–4, which aims to increase access to free, inclusive, and quality education, and ensure school-completion rates for all girls and boys by 2030 [4].

The study found that women who resided in wealthier households were less likely to use TBAs and other unskilled birth attendants. Consistent with this finding, past studies conducted in Nigeria [66], Zambia [53], and Ghana [64] showed the negative dose–response relationship between household wealth status and use of TBAs. A possible explanation for this observation could be that women who resided in poorer households may be constrained by finance and distance in accessing health facilities for birthing, and this may result in women opting for home birthing and subsequent use of TBAs or other unskilled birth attendants [52,53,67]. Furthermore, our analysis also showed that women who had money to pay for health services were less likely to use unskilled birth attendants during delivery. A recent study found that, in addition to household wealth status, having financial savings was a strong predictor of skilled-birth-attendant-assisted delivery [67]. This is because having access to financial savings may cover the considerable expenses associated with maternal-health-service utilization, including transportation, laboratory tests, and medications. Policy implications of this finding are that improving women’s financial power, having government schemes that match pregnancy-related savings, and the further subsidization of delivery services would be crucial to reduce the level of unskilled birth attendants’ delivery in Nigeria.

The present study showed that higher birth interval (≥2 years) was associated with the use of unskilled birth attendants compared to no previous birth, a finding that has been reported in similar studies conducted in Kenya [34] and Bangladesh [68]. The positive association between higher birth interval and the utilization of unskilled birth attendants may be explained by women having more confidence about receiving assistance from unskilled birth attendants after the first birth and may not seek skilled birth attendants in subsequent pregnancies. Previous studies conducted in Nigeria [69,70] and India [29] have shown that younger women were less likely to use ANC compared to older women, which may explain why those in the youngest age groups (15–24 years) were more likely to utilize unskilled birth attendants in the present study. Also, this study found that women who had frequent ANC (≥4) visits were less likely to be assisted by unskilled birth attendants compared to those who had no ANC visits. Women who previously had a poor experience with skilled birth attendants have also been shown to opt for home delivery and receive assistance from unskilled birth attendants, which may also contribute to this finding [71]. Improving ANC visits among all Nigerian women, especially among younger women, may have a positive impact on the mother’s decision to receive assistance from skilled birth attendants.

Research has indicated that media exposure improved women’s capacity to obtain, process, and understand basic health information, which is essential for making appropriate health-care decisions [72]. The current study showed that women who read magazines/newspapers or watched televisions had lower odds of being assisted by unskilled birth attendants. Previously published studies from Bangladesh [73], Ethiopia, and Eritrea [74] also found similar evidence that media exposure was associated with health-facility birthing with skilled birth attendants. Improving health literacy through the media may be a key national strategy to increase the proportion of women who give birth in health facilities with assistance from skilled health personnel in Nigeria.

### Study Limitations and Strengths

This study has several limitations. Firstly, we used the NDHS, which provides cross-sectional data, and therefore, prevents the assessment of the temporal relationship between the study factors and outcome measures. Secondly, the data may be subject to a degree of recall bias, given that responses were based on self-report, which may have resulted in measurement bias, and thus, led to an overestimation or underestimation of the effect size. Thirdly, the lack of assessment of unmeasured potential confounders (e.g., health-care-access data) is another limitation of the study. Fourthly, we were unable to separate trained TBAs from untrained TBAs in the NDHS, information that would have provided high-quality epidemiological data on both sets of TBAs, given that studies have reported some health benefits among trained TBAs [9,58]. Finally, there are no comprehensive data on the national or subnational maternal-health interventions during Nigeria’s longest democratic rule (1999–2018) to effectively examine the scope of the impact of these efforts on unskilled birth assistance, nor is there concurrent pre-democratic data to quantitatively correlate the implementation of a policy change in the use of unskilled birth attendants. Although there is not a formal relationship between policy implementation and the use of unskilled birth attendants over time, this study provides a detailed description of how likely the use of unskilled birth attendants has changed during the period of democratic governance in Nigeria. The study also examined the determinants of any change using large nationally representative surveys that span 19 years of democratization in Nigeria, including a weighted total of 103,903 participants. The NDHS is also conducted with trained personnel and standardized questionnaires that have been validated, lending validity to the data. Our study provides valuable insight into the long-term trends in the use of TBAs and other unskilled birth attendants amongst Nigerian women in the longest period of democratic governance in the country. The steady use of TBAs during this time underscores the need for health practitioners and policymakers to design initiatives that encourage facility birthing with skilled birth attendants.

## 5. Conclusions

Our study showed that the use of TBAs during delivery has remained relatively unchanged during the period of democratic governance in Nigeria between 1999 and 2018. In contrast, a decline in the proportion of women who utilized other unskilled birth attendants was evident from 2003 and 2018. Higher parental education, maternal employment, belonging to rich households, media exposure, higher maternal age (35–49 years), frequent ANC (≥4) visits, the proximity of health facilities, and women’s autonomy were associated with lower odds of unskilled-birth-attendant use in Nigeria. Rural residence, lower maternal age (15–24 years), and higher birth interval (≥2 years) were associated with higher odds of unskilled-birth-attendant use during delivery. These findings suggest that policy interventions that focus on improving girl-child education, health service access, media exposure, and socioeconomic opportunities for all women, particularly those who live in rural Nigeria, are urgently needed, to increase skilled-birth-attendant-assisted deliveries.

## Figures and Tables

**Figure 1 ijerph-17-00372-f001:**
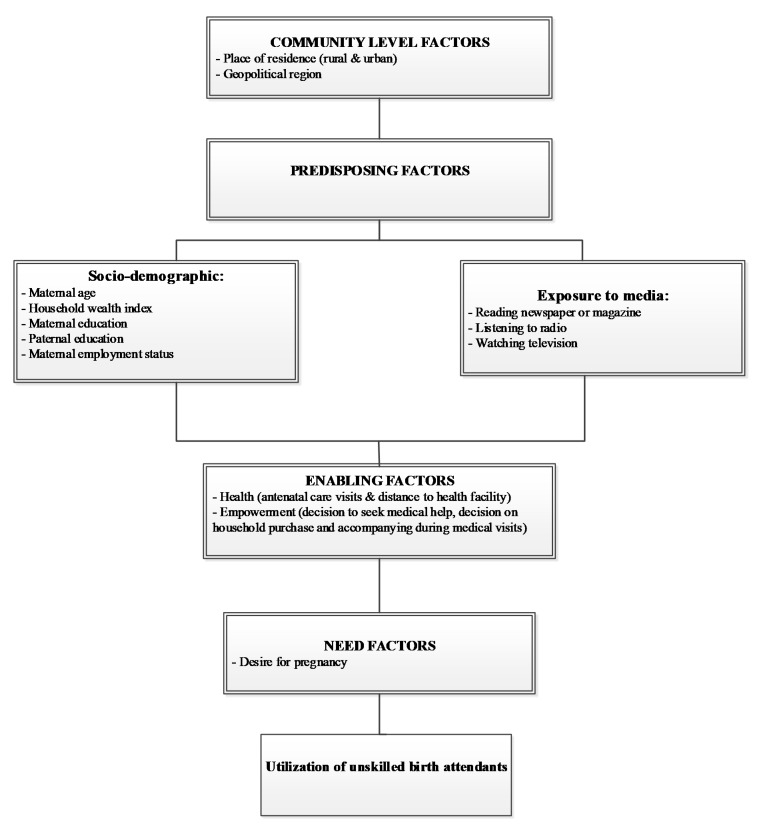
The conceptual model for the use of unskilled birth attendants in Nigeria, modified from Anderson’s health service utilization model [36].

**Figure 2 ijerph-17-00372-f002:**
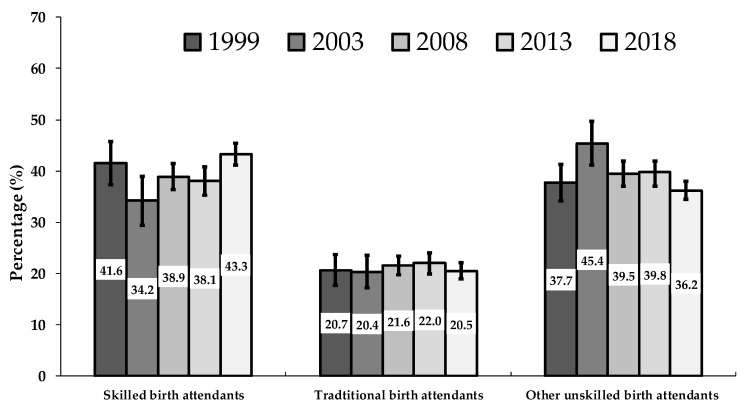
Trends in delivery assistance in Nigeria, 1999–2018. Error bars indicate 95% confidence interval.

**Table 1 ijerph-17-00372-t001:** Distribution of traditional birth attendant delivery by study factors in Nigeria, 1999–2018.

Variables	1999	2003	2008	2013	2018	1999–2018	1999–2018
	n (%)	n (%)	n (%)	n (%)	n (%)	n (%)	% Change (95% CI)
**Community-level factors**							
Place of residence							
Urban	153 (15.5)	208 (11.6)	1099 (13.1)	1320 (11.9)	1632 (12.4)	4412 (12.5)	−3.2 (−9.5, 3.21)
Rural	581 (22.7)	1061(24.0)	4970 (25.2)	5697 (27.5)	5374 (25.6)	17683 (25.8)	2.9 (−1.2, 7.1)
Geopolitical region							
North-Central	228 (28.9)	55 (6.1)	362 (9.5)	156 (3.6)	101 (2.2)	902 (6.2)	−26.7 (−34.8, −18.7)
Northeast	243 (38.7)	376 (25.5)	1539 (33.7)	1454 (26.1)	1010 (16.3)	4622 (25.0)	−22.4 (−30.2, −14.6)
Northwest	117 (18.1)	525 (24.3)	2273 (25.9)	4001 (34.0)	4565 (36.4)	11482 (32.0)	18.3 (11.9, 24.7)
Southeast	82 (10.6)	11 (3.0)	229 (8.4)	210 (7.4)	116 (3.4)	649 (6.4)	−7.2 (−10.9, −3.5)
Southwest	63 (9.0)	254 (32.2)	1205 (32.9)	874 (29.8)	876 (29.5)	3272 (29.6)	20.6 (15.2, 25.9)
South-South *		48 (9.0)	461 (10.2)	322 (7.4)	338 (7.7)	1168 (8.5)	
**Predisposing factors**							
**Sociodemographic factors**							
Maternal education							
No schooling	508 (29.6)	848 (26.3)	3650 (27.9)	4963 (31.7)	4571 (28.8)	14539 (29.4)	−0.8 (−6.1, 4.6)
Primary school	129 (14.8)	288 (19.6)	1474 (22.6)	1132 (18.5)	1142 (22.4)	4164 (20.7)	7.6 (3.3, 11.8)
Secondary and above	98 (10.1)	134 (8.7)	944 (11.1)	923 (9.2)	1293 (9.8)	3391 (9.9)	−0.4 (−3.5, 2.8)
Maternal employment							
No employment	453 (26.5)	476 (21.7)	2117 (24.9)	2653 (28.4)	2470 (25.3)	8168 (25.9)	−1.2 (−6.4, 4.0)
Formal employment	174 (14.5)	522 (19.3)	1927 (16.8)	2937 (19.4)	3713 (20.5)	9273 (19.1)	6.0 (2.4, 9.7)
Informal employment	25 (11.1)	70 (13.6)	1993 (25.0)	1324 (18.9)	192 (15.3)	3604 (21.2)	4.3 (−2.0, 10.5)
Partner education							
No schooling	422 (29.4)	648 (24.7)	3004 (27.8)	4190 (32.7)	3813 (28.1)	12077 (29.2)	−1.3 (−6.9, 4.2)
Primary school	128 (14.9)	328 (23.1)	1304 (21.5)	1226 (20.8)	1018 (22.5)	4005 (21.4)	7.6 (3.4, 11.8)
Secondary and above	182 (14.7)	282 (13.2)	1646 (15.3)	1526 (11.9)	1951 (12.5)	5588 (13.2)	−2.2 (−5.6, 1.2)
Household wealth status							
Poor	390 (27.5)	793 (28.6)	3550 (27.5)	4678 (31.5)	4483 (29.2)	13894 (29.4)	1.7 (−3.3, 6.6)
Middle	273 (18.2)	273 (21.7)	1436 (26.5)	1205 (20.1)	1325 (18.8)	4512 (21.3)	0.6 (−3.8, 5.0)
Rich	57 (10.0)	204 (9.3)	1082 (11.1)	1134 (10.3)	1197 (10.2)	3674 (10.4)	0.2 (−3.5, 3.8)
Maternal age							
25–34 years	325 (18.7)	611 (19.9)	2774 (19.8)	3277 (20.7)	3309 (19.0)	10296 (19.8)	0.3 (−3.1, 3.8)
15–24 years	274 (24.8)	408 (23.3)	1887 (27.1)	2063 (26.3)	1982 (24.3)	6613 (25.6)	−0.5 (−5.6, 4.6)
35–49 years	135 (19.1)	251 (17.9)	1407 (19.7)	1677 (20.6)	1714 (19.8)	5185 (19.9)	0.7 (−4.2, 5.6)
Birth interval							
No previous birth	157 (21.1)	241 (18.8)	1148 (21.4)	1275 (20.6)	1104 (16.7)	3923 (19.4)	−4.4 (−8.8, −0.0)
<24 months	121 (19.7)	275 (23.0)	1246 (23.1)	1334 (22.5)	1582 (23.1)	4558 (22.8)	3.4 (−1.4, 8.2)
≥24 months	455 (20.9)	752 (20.1)	3671 (21.2)	4402 (22.5)	4317 (20.9)	13598 (21.4)	0.0 (−3.7, 3.8)
**Exposure to the media**							
Listening radio *							
No		429 (24.0)	2841 (29.2)	3397 (27.0)	3653 (22.4)	10321 (25.5)	−1.5 (−7.2, 4.1) **
Yes		838 (19.0)	3194 (17.5)	3585 (18.8)	3352 (18.7)	10969 (18.4)	−0.3 (−4.1, 3.5) **
Read magazine or newspaper *							
No		1195 (22.9)	5716 (24.2)	6654 (24.4)	6678 (22.0)	20242 (23.4)	−0.8 (−4.8, 3.1) **
Yes		60 (6.3)	301 (7.1)	320 (7.3)	328 (8.5)	1009 (7.5)	2.2 (−0.8, 5.3) **
Watch television *							
No		965 (23.9)	4350 (26.7)	5172 (30.0)	5021 (25.7)	15508 (27.1)	1.7 (−3.0, 6.4) **
Yes		304 (13.9)	1686 (14.5)	1818 (12.6)	1985 (13.6)	5793 (13.5)	−0.3 (−4.0, 3.3) **
**Enabling factors**							
Antenatal Visit							
None	389 (32.7)	885 (23.5)	4277 (25.4)	5288 (28.8)	4483 (25.5)	15322 (26.5)	−7.3 (−12.7, 1.8)
1–3 visits	73 (19.4)	132 (24.8)	421 (24.8)	496 (20.1)	843 (22.4)	1965 (22.2)	3.0 (−2.6, 8.6)
≥4 visits	223 (13.3)	246 (25.2)	1074 (13.6)	1171 (11.2)	1642 (13.2)	4357 (12.7)	−0.1 (−3.2, 3.0)
Distance from health facility *							
A big problem		733 (25.5)	2763 (25.4)	3261 (32.2)	2109 (22.0)	8866 (26.5)	−3.6 (−8.9, 1.8) **
Not a big problem		533 (16.0)	3274 (19.1)	3727 (17.3)	4896 (19.9)	12432 (18.7)	3.9 (0.3, 7.6) **
Seek permission to visit health services *							
A big problem		400 (25.6)	1146 (25.8)	1339 (34.1)	844 (20.9)	3730 (26.7)	−4.7 (−11.0, 1.6) **
Not a big problem		866 (18.6)	4882 (20.8)	5647 (20.4)	6162 (20.4)	17557 (20.4)	1.8 (−1.6, 5.3)
Getting money to pay health services							
A big problem		841 (24.3)	4164 (25.1)	3611 (25.7)	3556 (21.4)	12172 (24.0)	−2.9 (−7.5, 1.6) **
Not a big problem		423 (15.4)	1882 (16.5)	3372 (19.2)	3450 (19.7)	9127 (18.5)	4.2 (0.5, 8.0) **
Accompany to health facility *							
A big problem		547 (27.9)	1366 (28.7)	1532 (33.1)	1334 (24.9)	4779 (28.6)	−3.0 (−9.6, 3.6) **
Not a big problem		715 (16.8)	4654 (20.1)	5454 (20.2)	5672 (19.7)	16494 (19.8)	2.8 (−0.5, 6.2) **
**Need factors**							
Desire for pregnancy							
Desired pregnancy	712 (21.3)	1200(20.5)	5771 (21.8)	6882 (22.2)	6870 (20.6)	21434 (21.4)	−0.7 (−4.3, 2.8)
Not desired pregnancy	17 (16.0)	69 (22.1)	245 (20.7)	62 (12.4)	136 (15.8)	530 (17.8)	−0.2 (−8.3, 7.8)

Notes: n (%) = weighted count and proportion for each outcome variable by study factors; * variables not reported in the 1999 Nigeria Demographic and Health Survey; ** % point change indicates percentage point change from 2003 to 2018.

**Table 2 ijerph-17-00372-t002:** Distribution of other-unskilled-birth-attendant deliveries by study factors in Nigeria, 1999–2018.

Variables	1999	2003	2008	2013	2018	1999–2018	1999–2018
	n (%)	n (%)	n (%)	n (%)	n (%)	n (%)	*% Change (95% CI)
**Community-level factors**							
Place of residence							
Urban	262 (26.6)	564 (31.4)	1795 (21.5)	2355 (21.2)	2633 (20.0)	7609 (21.5)	−6.6 (−13.0, −0.1)
Rural	1078 (42.0)	2259 (51.1)	9298 (47.1)	10314 (49.8)	9759 (46.4)	32709 (47.8)	4.4 (−0.4, 9.3)
Geopolitical region							
North-Central	461 (58.4)	412 (46.0)	1833 (47.9)	2165 (49.9)	2161 (46.8)	7031 (48.6)	−11.6 (−22.3, −0.9)
Northeast	335 (53.3)	816 (55.4)	2327 (50.9)	3014 (54.0)	3661 (58.9)	10153 (55.0)	5.6 (−2.0, 13.3)
Northwest	107 (16.6)	1370 (63.4)	5647 (64.3)	6322 (53.7)	5703 (45.4)	19148 (53.3)	28.9 (22.7, 35.0)
Southeast	126 (16.2)	43 (11.6)	267 (9.8)	295 (10.4)	392 (11.5)	1123 (11.1)	−4.8 (−9.7, 0.1)
Southwest	310 (43.9)	106 (13.4)	417 (11.4)	435 (14.8)	169 (5.7)	1438 (13.0)	−38.2 (−45.2, −31.2)
South-South *		76 (14.7)	602 (13.3)	439 (10.1)	306 (6.9)	1425 (10.3)	
**Predisposing factors**							
**Sociodemographic factors**							
Maternal education							
No schooling	952 (55.5)	1977 (61.3)	7914 (60.6)	8859 (56.6)	8999 (56.8)	28702 (58.0)	1.2 (−4.0, 6.5)
Primary school	256 (29.4)	553 (37.8)	2167 (33.2)	2282 (37.2)	1622 (31.8)	6880 (34.3)	2.4 (−2.4, 7.2)
Secondary and above	132 (13.7)	293 (19.1)	1012 (11.9)	1529 (15.2)	1772 (13.4)	4737 (13.8)	−0.3 (−3.4, 2.9)
Maternal employment							
No employment	863 (50.5)	1130 (51.5)	3939 (46.4)	3997 (42.8)	4487 (45.9)	14416 (45.7)	−4.6 (−10.0, 0.8)
Formal employment	293 (24.3)	1118 (41.4)	3785 (33.0)	5301 (35.0)	5171 (28.6)	15668 (32.2)	4.3 (0.1, 8.4)
Informal employment	53 (22.9)	252 (48.5)	3308 (41.4)	3225 (46.1)	305 (24.4)	7143 (42.1)	1.5 (−8.9, 12.0)
Partner education							
No schooling	772 (53.7)	1576 (60.0)	6465 (59.8)	7106 (55.4)	7381 (54.3)	23300 (56.4)	0.6 (−5.0, 6.2)
Primary school	289 (33.6)	616 (43.4)	2078 (34.3)	2266 (38.5)	1567 (34.6)	6816 (36.3)	1.0 (−4.3, 6.3)
Secondary and above	274 (22.1)	620 (29.2)	2311 (21.5)	3151 (24.6)	3312 (21.3)	9668 (22.8)	−0.9 (−5.1, 3.4)
Household wealth status							
Poor	693 (48.9)	1592 (57.4)	7707 (59.7)	8477 (57.1)	8275 (53.9)	26743 (56.5)	5.0 (−0.3, 10.3)
Middle	513 (34.3)	660 (52.6)	1948 (36.0)	2401 (40.0)	2506 (35.6)	8028 (37.8)	1.3 (−3.9, 6.5)
Rich	104 (18.2)	571 (26.1)	1438 (14.7)	1792 (16.3)	1621 (13.7)	5517 (15.6)	−4.6 (−10.4, 1.3)
Maternal age							
25–34 years	588 (33.8)	1312 (42.8)	5107 (36.5)	6036 (38.0)	5993 (34.5)	19036 (36.6)	0.7 (−3.6, 4.9)
15–24 years	471 (42.7)	844 (48.3)	2959 (42.5)	3286 (41.9)	3304 (40.6)	10865 (42.1)	−2.1 (−7.8, 3.5)
35–49 years	280 (39.6)	667 (47.5)	3027 (42.4)	3348 (41.2)	3096 (35.7)	10417 (40.0)	−3.9 (−9.3, 1.6)
Birth interval							
No previous birth	239 (32.2)	474 (37.1)	1584 (29.5)	1852 (29.9)	1805 (27.2)	5954 (29.5)	−5.0 (−10.0, 0.0)
<24 months	266 (43.4)	570 (47.7)	2280 (42.3)	2612 (44.0)	2668 (38.9)	8395 (42.0)	−4.4 (−10.7, 1.8)
≥24 months	833 (38.2)	1777 (47.5)	7227 (41.8)	8186 (41.8)	7911 (38.3)	25935 (40.9)	0.1 (−4.1, 4.3)
**Exposure to the media**							
Listening radio *							
No		970 (54.2)	4886 (50.2)	6514 (51.7)	8118 (49.8)	20487 (50.7)	−4.4 (−10.9, 2.2) **
Yes		1852 (42.1)	6161 (33.8)	6104 (32.0)	4275 (23.9)	18392 (30.8)	−18.2 (−23.2, −13.2) **
Read magazine or newspaper *							
No		2633 (50.4)	10640 (45.0)	11994 (44.0)	12081 (39.8)	37348 (43.2)	−10.6 (−15.2, 5.9) **
Yes		173 (18.0)	341 (8.1)	592 (13.5)	312 (8.1)	1418 (10.6)	−9.9 (−16.8, −3.1) **
Watch television *							
No		2256 (56.0)	8868 (54.3)	9219 (53.5)	10079 (51.5)	30422 (53.2)	−4.5 (−9.5, 0.6) **
Yes		566 (25.9)	2165 (18.6)	3386 (23.5)	2313 (15.8)	8430 (19.7)	−10.0 (−15.8, −4.3) **
**Enabling factors**							
Antenatal Visit							
None	751 (63.2)	2098 (55.8)	8510 (50.5)	8742 (47.6)	7893 (44.8)	27993 (48.4)	−18.4 (−23.8, −12.9)
1–3 visits	153 (40.9)	240 (45.1)	695 (40.9)	1257 (50.8)	1762 (46.9)	4106 (46.5)	5.9 (−1.1, 13.0)
4+ visits	360 (21.5)	467 (25.2)	1565 (19.9)	2557 (24.5)	2718 (21.8)	7667 (22.3)	0.4 (−3.4, 4.1)
Distance from health facility *							
A big problem		1495 (52.1)	5048 (46.5)	4704 (46.5)	4407 (45.9)	15654 (46.8)	−6.2 (−12.6, 0.2) **
Not a big problem		1323 (39.6)	5995 (35.0)	7907 (36.7)	7986 (32.5)	23210 (34.8)	−7.1 (−12.6, −1.7) **
Seek permission to visit health services *							
A big problem		856 (54.8)	2380 (53.5)	1983 (50.4)	1980 (49.1)	7199 (51.5)	−5.7 (−12.7, 1.3) **
Not a big problem		1964 (42.3)	8658 (36.8)	10609 (38.3)	10412 (34.5)	31645 (36.8)	−7.7 (−12.7, −2.8) **
Getting money to pay health services *							
A big problem		1683 (48.7)	6887 (41.6)	6222 (44.2)	7076 (42.5)	21869 (43.1)	−6.1 (−11.6, −0.7) **
Not a big problem		1130 (41.2)	4161 (36.4)	6371 (36.2)	5317 (30.3)	16978 (34.4)	−10.9 (−16.7, −5.1) **
Accompany to health facility *							
A big problem		1092 (55.7)	2371 (49.9)	2287 (49.4)	2367 (44.2)	8117 (48.6)	−11.5 (−18.2, −4.8) **
Not a big problem		1726 (40.7)	8647 (37.3)	10314 (38.1)	10026 (34.8)	30714 (36.9)	−5.9 (−11.0, −0.8) **
**Need factors**							
Desire for pregnancy							
Desired pregnancy	1232 (36.9)	2715 (46.3)	10419 (39.4)	12305 (39.7)	12185 (36.6)	38857 (38.9)	−0.3 (−4.4, 3.8)
Not desired pregnancy	18 (16.4)	71 (23.0)	294 (24.9)	123 (24.6)	208 (24.0)	715 (24.1)	7.6 (−1.8, 17.0)

Notes: n (%) = weighted count and proportion for each outcome variable by study factors; * variables not reported in the 1999 Nigeria Demographic and Health Survey; ** % point change indicates percentage point change from 2003 to 2018.

**Table 3 ijerph-17-00372-t003:** Determinants of traditional-birth-attendant-assisted delivery in Nigeria, 1999–2018.

Variables	1999	2003	2008	2013	2018	1999–2018	*p* for Trend
	aOR (95% CI)	aOR (95% CI)	aOR (95% CI)	aOR (95% CI)	aOR (95% CI)	aOR (95% CI)	
**Community-level factors**							
Place of residence							
Urban	1.00	1.00	1.00	1.00	1.00	1.00	0.412
Rural	0.92 (0.55–1.53)	1.51 (1.00–2.28)	1.48 (1.13–1.94)	1.80 (1.41–2.31)	1.31 (1.08–1.58)	1.46 (1.28–1.66)	0.034
Geopolitical region							
North-Central	1.00	1.00	1.00	1.00	1.00	1.00	<0.001
Northeast	1.60 (0.88–2.92)	7.20 (3.61–14.37)	8.40 (5.71–12.37)	11.56 (7.56–17.66)	6.24 (4.13–9.43)	7.66 (1.35–1.72)	<0.001
Northwest	0.70 (0.33–1.49)	14.61 (7.06–30.23)	14.16 (9.96–20.11)	22.36 (14.61–34.22)	21.21 (14.78–30.42)	16.47 (13.42–20.20)	0.687
Southeast	0.32 (0.16–0.62)	0.30 (0.10–0.86)	1.28 (0.83–1.98)	2.95 (1.74–5.00)	1.04 (0.63–1.74)	1.59 (1.20–2.10)	0.046
Southwest	0.16 (0.08–0.30)	8.52 (4.05–17.94)	8.77 (6.11–12.59)	18.96 (12.20–29.47)	20.10 (13.45–30.05)	12.83 (10.32–15.95)	<0.001
South-South *	0.00	4.71 (2.03–10.93)	2.35 (1.46–3.73)	3.18 (1.72–5.90)	4.81 (2.83–8.17)	2.93 (2.17–3.95) **	<0.001
**Predisposing factors**							
**Socioeconomic factors**							
Maternal education							
No schooling	1.00	1.00	1.00	1.00	1.00	1.00	0.003
Primary school	0.43 (0.26–0.72)	0.56 (0.36–0.88)	0.67 (0.56–0.81)	0.61 (0.50–0.75)	0.76 (0.61–0.95)	0.66 (0.59–0.74)	0.037
Secondary or higher	0.25 (0.15–0.42)	0.24 (0.14–0.42)	0.37 (0.29–0.47)	0.36 (0.28–0.47)	0.39 (0.31–0.49)	0.37 (0.32–0.42)	<0.001
Maternal employment							
No employment	1.00	1.00	1.00	1.00	1.00	1.00	0.009
Formal employment	0.86 (0.60–1.23)	0.97 (0.69–1.36)	0.74 (0.63–0.87)	0.94 (0.78–1.13)	0.89 (0.78–1.02)	0.87 (0.79–0.95)	0.023
Informal employment	0.53 (0.30–0.94)	0.89 (0.53–1.52)	0.89 (0.73–1.08)	0.92 (0.74–1.15)	0.70 (0.49–1.01)	0.93 (0.81–1.05)	0.005
Partner education							
No schooling	1.00	1.00	1.00	1.00	1.00	1.00	0.023
Primary school	0.39 (0.25–0.60)	1.19 (0.78–1.80)	0.67 (0.56–0.81)	0.69 (0.57–0.83)	0.79 (0.65–0.96)	0.74 (0.66––0.82)	0.253
Secondary or higher	0.56 (0.36–0.85)	0.73 (0.49–1.08)	0.62 (0.51–0.76)	0.55 (0.46–0.66)	0.65 (0.55–0.77)	0.62 (0.56–0.69)	0.430
Household wealth status							
Poor	1.00	1.00	1.00	1.00	1.00	1.00	0.020
Middle	0.60 (0.41–0.87)	0.62 (0.41–0.93)	0.74(0.60–0.90)	0.64 (0.52–0.78)	0.51 (0.42–0.62)	0.59 (0.53–0.67)	<0.001
Rich	0.34 (0.19–0.61)	0.21 (0.12–0.34)	0.36 (0.27–0.46)	0.49 (0.38–0.64)	0.37 (0.30–0.46)	0.37 (0.33–0.43)	0.001
Maternal age							
25–34 years	1.00	1.00	1.00	1.00	1.00	1.00	0.410
15–24 years	1.90 (1.30–2.80)	1.60 (1.09–2.35)	1.35 (1.17–1.57)	1.32 (1.12–1.56)	1.49 (1.28–1.73))	1.41 (1.29–1.54)	0.072
35–49 years	0.62 (0.40–0.95)	0.49 (0.31–0.78)	0.90 (0.78–1.03)	0.77 (0.66–0.90)	0.93 (0.80–1.07)	0.87 (0.81–0.94)	0.062
Birth interval							
No previous birth	1.00	1.00	1.00	1.00	1.00	1.00	0.105
<24 months	1.60 (0.99–2.6)	1.65 (1.07–2.53)	1.55 (1.32–1.82)	1.39 (1.17–1.66)	1.87 (1.59–2.19)	1.60 (1.45–1.76)	0.458
≥24 months	1.89 (1.26–2.81)	1.67 (1.12–2.49)	1.44 (1.25–1.65)	1.60 (1.38–1.84)	1.82 (1.60–2.07)	1.62 (1.50–1.75)	0.054
**Exposure to the media**							
Listening radio *							
No		1.00	1.00	1.00	1.00	1.00	<0.001
Yes		1.18 (0.80–1.76)	0.76 (0.65–0.89)	1.15 (0.97–1.35)	1.02 (0.88–1.19)	0.96 (0.88–1.05) **	<0.001
Read magazine or newspaper *							
No		1.00	1.00	1.00	1.00	1.00	0.025
Yes		0.65 (0.37–1.14)	0.62 (0.50–0.78)	0.66 (0.54–0.82)	0.70 (0.56–0.88)	0.65 (0.58–0.74) **	0.043
Watch television *							
No		1.00	1.00	1.00	1.00	1.00	0.130
Yes		1.40 (0.89–2.21)	0.98 (0.82–1.18)	0.71 (0.59–0.84)	0.80 (0.68–0.94)	0.86 (0.78–0.95) **	0.279
**Enabling factors**							
Antenatal Visit							
None	1.00	1.00	1.00	1.00	1.00	1.00	0.538
1–3 visits	0.18 (0.11–0.32)	0.89 (0.56–1.41)	0.83 (0.69–1.00)	0.66 (0.55–0.79)	0.76 (0.65–0.88)	0.74 (0.67–0.82)	0.308
≥4 visits	0.10 (0.06–0.17)	0.46 (0.35–0.61)	0.46 (0.41–0.52)	0.40 (0.36–0.45)	0.45 (0.41–0.50)	0.44 (0.41–0.47)	0.860
Distance from health facility *							
A big problem		1.00	1.00	1.00	1.00	1.00	0.125
Not a big problem		0.76 (0.50–1.15)	0.94 (0.78–1.12)	0.61 (0.50–0.74)	0.86 (0.70–1.07)	0.77 (0.69–0.86) **	0.008
Seek permission to visit health services *							
A big problem		1.00	1.00	1.00	1.00	1.00	0.684
Not a big problem		1.15 (0.77–1.71)	0.90 (0.72–1.09)	0.81 (0.62–1.05)	1.14 (0.89–1.47)	0.96 (0.84–1.09) **	0.465
Getting money to pay health services *							
A big problem		1.00	1.00	1.00	1.00	1.00	0.030
Not a big problem		0.86 (0.61–1.22)	0.69 (0.59–0.82)	0.82 (0.69–0.97)	0.96 (0.80–1.14)	0.85 (0.78–0.94) **	0.239
Accompany to health facility*							
A big problem		1.00	1.00	1.00	1.00	1.00	0.163
Not a big problem		0.68 (0.44–1.04)	0.67 (0.56–0.80)	0.82 (0.66–1.03)	0.74 (0.59–0.93)	0.78 (0.69–0.87) **	0.421
**Need factors**							
Desire for pregnancy							
Desired pregnancy	1.00	1.00	1.00	1.00	1.00	1.00	0.126
Not desired pregnancy	0.53 (0.21–1.34)	0.45 (0.22–0.93)	1.20 (0.92–1.58)	0.81 (0.57–1.16)	0.81 (0.57–1.14)	0.87 (0.71–1.06)	0.055

Notes: aOR = adjusted odds ratio; * variables not reported in the 1999 Nigeria Demographic and Health Survey; in the model of community-level factors, adjustments were conducted for predisposing (sociodemographic and media exposure), enabling, and need factors. Similar approaches were used for the predisposing, enabling, and need factors, with adjustments for respective factors in multivariate models; ** analyses based on the 2003–2018 Nigeria Demographic and Health Survey dataset.

**Table 4 ijerph-17-00372-t004:** Determinants of other-unskilled-birth-attendant-assisted delivery in Nigeria, 1999–2018.

Variables	1999	2003	2008	2013	2018	1999–2018	*p* for Trend
	aOR (95% CI)	aOR (95% CI)	aOR (95% CI)	aOR (95% CI)	aOR (95% CI)	aOR (95% CI)	
**Community-level factors**							
Place of residence							
Urban	1.00	1.00	1.00	1.00	1.00	1.00	<0.001
Rural	1.53 (1.02–2.30)	1.50 (0.99–2.26)	1.43 (1.36–1.80)	1.54 (1.23–1.92)	1.42 (1.18- 1.71)	1.51 (1.34- 1.71)	<0.001
Geopolitical region							
North-Central	1.00	1.00	1.00	1.00	1.00	1.00	0.043
Northeast	1.09 (0.58–2.03)	1.91 (1.18–3.09)	1.74 (1.30–2.34)	1.86 (1.41–2.45)	1.55 (1.20–1.99)	1.71 (1.46–1.99)	0.026
Northwest	0.23 (0.10–0.51)	4.92 (3.03–7.98)	4.36 (3.31–5.75)	2.92 (2.14–3.97)	1.84 (1.45–2.34)	2.78 (2.38–3.26)	<0.001
Southeast	0.20 (0.11–0.38)	0.09 (0.04–0.20)	0.21 (0.14–0.32)	0.23 (0.16–0.33)	0.31 (0.21–0.45)	0.23 (0.19–0.29)	0.036
Southwest	0.47 (0.27–0.83)	0.45 (0.24–0.82)	0.47 (0.33–0.65)	0.61 (0.41–0.89)	0.26 (0.18–0.39)	0.46 (0.37–0.58)	0.588
South-South *		0.62 (0.36–1.20)	0.46 (0.33–0.65)	0.29 (0.19–0.45)	0.28 (0.20–0.39)	0.35 (0.28–0.43) **	<0.001
**Predisposing factors**							
**Socioeconomic factors**							
Maternal education							
No schooling	1.00	1.00	1.00	1.00	1.00	1.00	<0.001
Primary school	0.64 (0.44–0.94)	0.68 (0.46–1.01)	0.72 (0.61–0.84)	0.69 (0.58–0.83)	0.56 (0.46–0.67)	0.66 (0.59–0.73)	<0.001
Secondary or higher	0.36 (0.23–0.57)	0.41 (0.26–0.65)	0.40 (0.33–0.50)	0.40 (0.31–0.51)	0.30 (0.25–0.36)	0.34 (0.31–0.39)	<0.001
Maternal employment							
No employment	1.00	1.00	1.00	1.00	1.00	1.00	<0.001
Formal employment	0.82 (0.63–1.06)	0.83 (0.64–1.08)	0.79 (0.67–0.92)	1.03 (0.86–1.23)	0.74 (0.65–0.83)	0.82 (0.75–0.89)	<0.001
Informal employment	1.09 (0.72–1.65)	1.13 (0.75–1.71)	0.84 (0.71–1.00)	1.24 (1.01–1.53)	0.68 (0.47–0.97)	1.03 (0.92–1.15)	0.001
Partner education							
No schooling	1.00	1.00	1.00	1.00	1.00	1.00	<0.001
Primary school	0.97 (0.68–1.40)	0.98 (0.69–1.40)	0.62 (0.53–0.73)	0.80 (0.67–0.96)	0.73 (0.61–0.88)	0.72 (0.66–0.80)	<0.001
Secondary or higher	0.61 (0.43–0.85)	0.62 (0.44–0.88)	0.58 (0.49–0.68)	0.64 (0.53–0.77)	0.66 (0.57–0.78)	0.61 (0.56–0.67)	<0.001
Household wealth status							
Poor	1.00	1.00	1.00	1.00	1.00	1.00	<0.001
Middle	0.88 (0.60–1.29)	0.88 (0.60–1.28)	0.49 (0.40–0.59)	0.56 (0.46–0.67)	0.69 (0.59–0.82)	0.60 (0.54–0.67)	<0.001
Rich	0.29 (0.19–0.46)	0.34 (0.21–0.54)	0.27 (0.22–0.35)	0.35 (0.27–0.45)	0.51 (0.41–0.62)	0.39 (0.35–0.45)	<0.001
Maternal age							
25–34 years	1.00	1.00	1.00	1.00	1.00	1.00	<0.001
15–24 years	1.89 (1.31–2.71)	1.55 (1.10–2.19)	1.19 (1.03–1.37)	1.21 (1.04–1.42)	1.40 (1.22–1.61)	1.30 (1.20–1.41)	<0.001
35–49 years	0.811 (0.56–1.16)	1.12 (0.81–1.56)	1.02 (0.90–1.15)	0.91 (0.80–1.05)	0.95 (0.84–1.08)	0.98 (0.91–1.05)	<0.001
Birth interval							
No previous birth	1.00	1.00	1.00	1.00	1.00	1.00	<0.001
<24 months	1.74 (1.23–2.47)	1.73 (1.21–2.46)	1.87 (1.59–2.18)	1.82 (1.52–2.18)	2.10 (1.81–2.4)	1.90 (1.73–2.07)	<0.001
≥24 months	2.10 (1.50–2.93)	2.04 (1.44–2.89)	1.81 (1.58–2.06)	1.92 (1.67–2.20)	2.21 (1.94–2.51)	1.97 (1.83–2.13)	<0.001
**Exposure to the media**							
Listening radio *							
No		1.00	1.00	1.00	1.00	1.00	0.003
Yes		1.03 (0.73–1.47)	0.97 (0.84–1.11)	1.00 (0.86–1.16)	0.73 (0.63–0.84)	0.94 (0.86–1.02) **	<0.001
Read magazine or newspaper *							
No		1.00	1.00	1.00	1.00	1.00	<0.001
Yes		0.84 (0.52–1.36)	0.50 (0.41–0.62)	0.73 (0.60–0.89)	0.55 (0.43–0.70)	0.67 (0.60–0.76) **	<0.001
Watch television *							
No		1.00	1.00	1.00	1.00	1.00	<0.001
Yes		0.92 (0.62–1.36)	0.87 (0.74–1.02)	0.82 (0.70–0.97)	0.65 (0.56–0.76)	0.77 (0.71–0.84) **	<0.001
**Enabling factors**							
Antenatal Visit							
None	1.00	1.00	1.00	1.00	1.00	1.00	<0.001
1–3 visits	0.59 (0.38–0.93)	0.59 (0.38–0.94)	0.72 (0.61–0.85)	0.77 (0.65–0.91)	0.76 (0.65–0.88)	0.70 (0.64–0.77)	0.003
≥4 visits	0.32 (0.25–0.42)	0.34 (0.26–0.44)	0.39 (0.35–0.44)	0.42 (0.38–0.46)	0.40 (0.36–0.44)	0.39 (0.36–0.41)	<0.001
Distance from health facility *							
A big problem		1.00	1.00	1.00	1.00	1.00	<0.001
Not a big problem		0.73 (0.50–1.08)	0.74 (0.63–0.86)	0.86 (0.71–1.04)	0.86 (0.71–1.05)	0.80 (0.72–0.88) **	<0.001
Seek permission to visit health services *							
A big problem		1.00	1.00	1.00	1.00	1.00	<0.001
Not a big problem		1.20 (0.83–1.73)	0.77 (0.63–0.93)	0.74 (0.58–0.96)	0.77 (0.62–0.97)	0.82 (0.73–0.93) **	<0.001
Getting money to pay health services *							
A big problem		1.00	1.00	1.00	1.00	1.00	<0.001
Not a big problem		1.00 (0.73–1.35)	1.07 (0.93–1.23)	0.80 (0.69–0.94)	0.84 (0.72–0.98)	0.91 (0.83–0.99) **	<0.001
Accompany to health facility *							
A big problem		1.00	1.00	1.00	1.00	1.00	<0.001
Not a big problem		0.69 (0.47–1.02)	0.81 (0.69–0.95)	0.90 (0.71–1.13)	1.21 (0.95–1.53)	0.91 (0.82–1.02) **	<0.001
**Need factors**							
Desire for pregnancy							
Desired the pregnancy	1.00	1.00	1.00	1.00	1.00	1.00	<0.001
Not desired pregnancy	0.49 (0.23–1.04)	0.48 (0.23–1.02)	1.40 (1.10–1.77)	1.02 (0.72–1.44)	0.95 (0.68–1.32)	1.06 (0.90–1.25)	0.080

Notes: aOR: = adjusted odds ratio; * variables not reported in the 1999 Nigeria Demographic and Health Survey; In the model of community-level factors, adjustments were conducted for predisposing (sociodemographic and media exposure), enabling, and need factors. Similar approaches were used for the predisposing, enabling, and need factors, with adjustments for respective factors in multivariate models; ** analyses based on the 2003–2018 Nigeria Demographic and Health Survey dataset.

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
