# Peer review of "Prevalence, Trends, and Drivers of the Utilization of Unskilled Birth Attendants during Democratic Governance in Nigeria from 1999 to 2018"

_ijerph, 2020, doi:10.3390/ijerph17010372_

Round 1

Reviewer 1 Report

The overall aim of this paper was to examine the prevalence, trends, and drivers of the utilization of unskilled birth attendants in Nigeria for a 19 year time period. There are a few comments for the authors: 

-Is the North Central region of Nigeria used as the reference since it is the most urban? As someone who is unfamiliar with Nigeria's geographic regions, I was not sure why it was chosen as the reference. 

-Statistical Analysis: Did the authors consider using JoinPoint regression analysis to calculate the percent change between the years? 

-Was there a reason the maternal age group from 15-24 years was chosen as the reference group? Often, it is the age group from 25-34 years that is chosen as the reference group. 

Overall, this was an interesting paper. 

Reviewer 2 Report

The topic is very relevant and illustrates the trends of use of TBA'S and unskilled Birth Attendants as well as the potential contributing factors. In general the paper is very well written.

I only have a few minor comments:

Trends are presented in Figure 2- was a trend analyses conducted to assess if the change was significant or not. E.g. the trend from 2003 to 2018 for the use of unskilled birth attendants (45.5% to 36.2%). Was this decrease significant?

Page 10- the odds are mention in the paragraph – Determinants of TBA assisted delivery. The actual adjusted odds ratios (numbers) must be stated as well with the 95% confidence intervals. The same applies to the paragraph on page 16.

In table 3 – not clear why (within each variable) a particular category was chosen as the reference category- e.g. in maternal education – no education which has the biggest % use of TBA is the reference – I would have preferred the lowest use of TBA or unskilled to the reference for each category. It will not change the outcomes but makes the table more clear.
